# Residual Motion Error Correction with Backprojection Multisquint Algorithm for Airborne Synthetic Aperture Radar Interferometry

**DOI:** 10.3390/s19102342

**Published:** 2019-05-21

**Authors:** Pengfei Xie, Man Zhang, Lei Zhang, Guanyong Wang

**Affiliations:** 1National Laboratory of Radar Signal Processing, Xidian University, Xi’an 710071, China; PerfeiShieh@126.com; 2School of Computer Science and Technology, Xidian University, Xi’an 710071, China; 3School of Electronics and Communication Engineering, Sun Yat-Sen University, Guangzhou 510275, China; 4Beijing Institute of Radio Measurement, The Second Academy of China Aerospace Science and Industry Corporation (CASIC), Beijing 100854, China; gywang.sar@gmail.com

**Keywords:** interferometric synthetic aperture radar (InSAR), backprojection (BP), multisquint (MSQ), residual motion error (RME), motion compensation (MoCo)

## Abstract

For airborne interferometric synthetic aperture radar (InSAR) data processing, it is essential to achieve precise motion compensation to obtain high-quality digital elevation models (DEMs). In this paper, a novel InSAR motion compensation method is developed, which combines the backprojection (BP) focusing and the multisquint (MSQ) technique. The algorithm is two-fold. For SAR image focusing, BP algorithm is applied to fully use the navigation information. Additionally, an explicit mathematical expression of residual motion error (RME) in the BP image is derived, which paves a way to integrating the MSQ algorithm in the azimuth spatial wavenumber domain for a refined RME correction. It is revealed that the proposed backprojection multisquint (BP-MSQ) algorithm exploits the motion error correction advantages of BP and MSQ simultaneously, which leads to significant improvements of InSAR image quality. Simulation and real data experiments are employed to illustrate the effectiveness of the proposed algorithm.

## 1. Introduction

Interferometric synthetic aperture radar (InSAR) is a bistatic radar system for remote sensing observation and digital elevation models (DEMs) acquisition. Compared with the spaceborne InSAR, airborne InSAR has the advantages of simple assembly, convenient maintenance, high flexibility and easy revisiting [1,2,3]. However, the relative position of the coherent antenna will change due to atmosphere turbulence and inertial navigation error in flight. Relative motion errors caused by a nonideal trajectory will dramatically contaminate interferometric phase of InSAR image [4,5,6], especially for high working frequency (e.g., millimeter-wave). As a result, high-precision phase preserving and refined motion error compensation in SAR focusing is necessary to generate precise DEM.

Conventional Doppler frequency domain imaging algorithms (like range-Doppler and chirp-scaling algorithm) are commonly used in airborne InSAR for image focusing. Motion compensation (MoCo) is also embedded into the imaging procedure with necessary approximations [7]. Supported by the global navigation satellite system (GNSS) and inertial measurement unit (IMU), airborne InSAR is able to record the trajectory information, so the interferometric phase error of multichannel images is reduced by MoCo. Limited by the GNSS/IMU accuracy and compensation ability, conventional motion compensated InSAR image pairs contain residual motion errors (RME). For example, the baseline time-varying errors will degrade the interferogram precision. Furthermore, it is difficult to combine accurate motion compensation with frequency domain imaging method when there are significant motion errors [8]. It is widely accepted that backprojection (BP) SAR interferometry is less sensitive to the interferometric baseline length and angle errors. Hence, the time domain BP algorithm is usually considered an ideal solver to SAR focusing and motion compensation. Compared with the frequency domain focusing algorithms, BP algorithm is more direct, flexible and precise in motion compensation [9,10]. Its implementation of point-by-point coherent accumulation exploits the recorded navigation information completely, yielding both focus and geometric preservation. Another advantage of the BP algorithm is obvious that the image coordinate is customizable so that the image can be mapped directly in the geographic coordinate, which is useful for InSAR image registration [11].

In the case of inaccuracy for available navigation information, there are many approaches to estimate RME, such as autofocusing methods [12,13], wavelet decomposition and polynomial fitting methods [14], and a widely applied multisquint (MSQ) technique as well as its improved methods [15,16,17,18]. Among those approaches, the MSQ algorithm is most wildly used. MSQ estimates RME in the image domain by generating interferograms with different oblique angles in different subapertures [19]. Conventional MSQ is a useful assistance tool with a high processing speed and precision, which usually combines with the frequency domain focusing process for InSAR RME estimation. A feasible combination of BP and MSQ was introduced in [2], in which RME was investigated by segmenting the BP InSAR image into subapertures in the azimuth time domain. However, the original MSQ application combining with BP algorithm is still limited. Obtaining RME by differential interferometric phase integration means that the algorithm is sensitive to high-frequency time-varying baseline errors and inapplicable to linear RME. Weighting least square estimation strategy would be helpful to improve its performance in linear RME estimation [20]. However, shortcomings in dealing with multiple SAR modes and different motion error properties prevent conventional MSQ in high frequency-band airborne InSAR applications. In this paper, a novel azimuth time domain MSQ algorithm is established, which seamlessly combines with BP focusing process to precisely compensate InSAR RME. The main contributions of this work are concluded as follows:(1)The BP algorithm is firstly used to accurately focus InSAR image pairs, then the subapertures (also called sublooks in this paper) for MSQ differential interferometry is evenly split in the azimuth wavenumber domain. In contrast to the process of the subaperture differential interferometric phase integration in [2], the proposed BP-MSQ algorithm estimates polynomial parameters of RME by using the subaperture differential phase, while it analyzes the representation of RME in the azimuth time domain. In this manner, both high- and low-frequency error components of RME can be obtained precisely.(2)The second contribution of this work is that accurate analytic expressions of RME in BP InSAR image pairs from stripmap and spotlight SAR modes are derived separately. In addition, the characteristics of subaperture differential interferometry phase diagrams with linear and high-order types RME are analyzed, which makes the RME estimation more flexible and adjustable to multiple imaging modes.(3)In the framework of the improved BP-MSQ algorithm, linear RME estimation flowcharts for stripmap SAR and spotlight SAR are given differently. Meanwhile, a piecewise RME model is developed for the refined high-order RME compensation under spotlight mode. Avoiding the differential phase integration in RME estimation, the developed BP-MSQ outperforms the original MSQ in the case of strong noises.

The paper is organized as follows. The mathematical model for the BP algorithm of InSAR with RME is introduced in Section 2. In Section 3, the principle for RME estimation with BP algorithm and MSQ technique is derived, the method for RME inversion in stripmap SAR and spotlight SAR is established, and the flowchart for RME estimation with BP-MSQ algorithm is described in detail. In Section 4, a series of simulation and real measured data experiments are given to illustrate the effectiveness of proposed algorithm. Finally, in Section 5 some conclusions are given for the RME estimation method proposed in this paper.

## 2. Airborne InSAR Focusing with Backprojection Imaging

The schematic diagram of InSAR system is shown in Figure 1. The system is bistatic, in which A is the master antenna, which is responsible for transmitting and receiving radar signals. B is the slave antenna, which is only responsible for receiving echo data. The baseline length between antenna phase center of A and *B* is L, and the baseline inclination angle (relative to the horizontal plane) is α. $T_{r}$ is the motion track of the airborne SAR, and the flight altitude of the aircraft is H. The elevation of target point P is h, and Q is the focal position of P on the reference horizontal plane. For airborne InSAR, the motion trajectory of the carrier is usually a nonideal straight line, and the motion error needs to be compensated to obtain better imaging results. For the purpose of accurate motion error compensation, the time domain BP algorithm is applied in this paper.

Generally, airborne SAR transmits a wideband pulse chirp (linear frequency modulation) signal, and achieves range focusing. For simplicity, taking the echo signal of the scattering point P as an example, the signal after the pulse compression can be given by [21], which is shown as follows
(1)$$s_{i}^{c}=C_{i}\exp\left\{-j2\pi f_{c}\tau_{i}\right\}$$
where *i* is the slow-time pulse index, $C_{i}$ is the amplitude after pulse compression, $f_{c}$ is the radar center frequency, and $\tau_{i}$ is the two-way propagation time from the radar antenna to *P*. The realization process of BP can be written by [2]
(2)$$\begin{array}{cc} I & =\sum_{i=-N}^{N}s_{i}^{c}\exp\left\{j2\pi f_{c}\tilde{\tau_{i}}\right\}\\ & =\sum_{i=-N}^{N}C_{i}\exp\left\{-j2\pi f_{c}\left(\tau_{i}-\tilde{\tau_{i}}\right)\right\} \end{array}$$
where *I* is defined as the final focus of the scattering point, *N* is the pulse index range of the synthetic aperture, and $\tilde{\tau_{i}}$ is the delay of the projection point *Q* of the target *P* and the following relationships exist.
(3)$$\begin{array}{c} \tau_{i}^{A}=\frac{2R_{APi}}{c}\\ \tau_{i}^{B}=\frac{\left(R_{APi}+R_{BPi}\right)}{c}\\ \tilde{\tau_{i}^{A}}=\frac{2R_{AQi}}{c}\\ \tilde{\tau_{i}^{B}}=\frac{\left(R_{AQi}+R_{BQi}\right)}{c} \end{array}$$

In Equation (Equation 3), the superscript *A* indicates the antenna *A* and the superscript *B* indicates the antenna *B*. $R_{APi}$ is the instantaneous slant range from the antenna *A* to the scattering point *P*, $R_{BPi}$ is the instantaneous slant range from the antenna *B* to the scattering point *P*, $R_{AQi}$ is the instantaneous slant range from antenna *A* to projection point *Q*, $R_{BQi}$ is the instantaneous slant range from antenna *B* to projection point *Q*, and *c* is the speed of light. In the actual imaging process, the instantaneous slant range is calculated using the measurement positions $A^{{}^{\prime }}$ and $B^{{}^{\prime }}$ of the antenna *A* and the antenna *B* to compensate the phase. However, since there is a slight measurement error between the measurement positions $A^{{}^{\prime }}$ and $B^{{}^{\prime }}$ with the actual positions *A* and *B*, the $\tilde{\tau_{i}^{A}}$ and $\tilde{\tau_{i}^{B}}$ in Equation (Equation 3) is revised as
(4)$$\begin{array}{cc} \hat{\tilde{\tau_{i}^{A}}} & =\frac{2R_{A^{\prime }Qi}}{c}\\ & =\frac{2R_{AQi}}{c}+\Delta \tau_{i}^{A}\\ \hat{\tilde{\tau_{i}^{B}}} & =\frac{\left(R_{A^{\prime }Qi}+R_{B^{\prime }Qi}\right)}{c}\\ & =\frac{\left(R_{AQi}+R_{BQi}\right)}{c}+\Delta \tau_{i}^{B} \end{array}$$
where $\Delta \tau_{i}^{A}$ and $\Delta \tau_{i}^{B}$ are the instantaneous delay deviations caused by the measurement errors of *A* and *B*. For simplify, it is assumed that the backscattering coefficients of the scattering point *P* to *A* and *B* are the same. Hence, the actual imaging process for *P* is given by
(5)$$\begin{array}{cc} I^{A} & =\sum_{i=-N}^{N}C_{i}^{A}\exp\left\{-j2\pi f_{c}\left(\tau_{i}^{A}-\tilde{\tau_{i}^{A}}\right)\right\}\exp\left\{j2\pi f_{c}\Delta \tau_{i}^{A}\right\}\\ I^{B} & =\sum_{i=-N}^{N}C_{i}^{B}\exp\left\{-j2\pi f_{c}\left(\tau_{i}^{B}-\tilde{\tau_{i}^{B}}\right)\right\}\exp\left\{j2\pi f_{c}\Delta \tau_{i}^{B}\right\} \end{array}$$
where the superscript *A* indicates the antenna *A* and the superscript *B* indicates the antenna *B*. The interferogram of the imaging result in Equation (Equation 5) can be obtained.
(6)$$\langle I^{A}\cdot I^{B*}\rangle =\sum_{i=-N}^{N}\tilde{C_{i}}\exp\left\{j\varphi_{i}\right\}\exp\left\{j\phi_{i}\right\}$$
where $\tilde{C_{i}}=C_{i}^{A}\cdot C_{i}^{B}$ is the amplitude term, and the phase term, $\varphi_{i}$ and $\phi_{i}$, respectively defined by
(7)$$\begin{array}{c} \varphi_{i}=-\frac{2\pi f_{c}}{c}\left(R_{APi}-R_{AQi}-R_{BPi}+R_{BQi}\right)\\ \phi_{i}=2\pi f_{c}\left(\Delta \tau_{i}^{A}-\Delta \tau_{i}^{B}\right) \end{array}$$
where $\varphi_{i}$ represents the instantaneous interferometric phase after flat-earth phase removal of the scattering point *P*, $\phi_{i}$ represents the interferometric phase error caused by the RME. After BP imaging using the track information, the RME caused by the measurement error is mainly reflected in the interferometric phase, i.e., the differential phase. An important way to correct InSAR RME is to extract relative motion error from differential phase and compensate it. For this way, MSQ technique is a popular and effective method to estimate RME from interferometric phase.

## 3. RME Estimation Principle with the Polynomial Fitting MSQ

### 3.1. RME Estimation Principle with the BP-MSQ Algorithm

Multisquint technique originated from spectral segmentation technology, which has been successfully used in RME estimation for InSAR. Firstly, subaperture images and corresponding subaperture interferogram can be obtained by segmenting the image signal in the frequency domain (azimuth Doppler domain). Then the interferograms of different subapertures are conjugate multiplied to obtain the differential phase of the RME. For the SAR system, the instantaneous Doppler of the target point corresponds to the squint angle. The SAR image signal is evenly divided in the azimuth Doppler domain, which essentially divides the SAR image signal into multiple subapertures in slow-time domain. Assuming that each subaperture contains *n* pulses, for two adjacent subapertures, the interferogram obtained by BP algorithm in each subaperture can be written as
(8)$$\begin{array}{c} {\langle I^{A}\cdot I^{B*}\rangle }_{m}=\sum_{i=s_{m}}^{s_{m}+n}\tilde{C_{i}}\exp\left\{j\varphi_{i}\right\}\exp\left\{j\phi_{i}\right\}\\ {\langle I^{A}\cdot I^{B*}\rangle }_{m+1}=\sum_{i=s_{m}+n}^{s_{m}+2n}\tilde{C_{i}}\exp\left\{j\varphi_{i}\right\}\exp\left\{j\phi_{i}\right\} \end{array}$$
where subscripts *m* and *m* + 1 indicate two adjacent subapertures, and $s_{m}$ is the starting pulse index of the subaperture *m*. The differential interferometric signal can be obtained by differentiating the interferogram of the two subapertures in Equation (Equation 8).
(9)$$\begin{array}{cc} {\langle I^{A}\cdot I^{B*}\rangle }_{m,m+1} & ={\langle I^{A}\cdot I^{B*}\rangle }_{m}\cdot {\langle I^{A}\cdot I^{B*}\rangle }_{m+1}^{*}\\ & =\sum_{i=s_{m}}^{s_{m}+n}\tilde{C_{i}}\tilde{C_{i+n}}\exp\left\{j\left(\varphi_{i}-\varphi_{i+n}\right)\right\}\exp\left\{j\left(\phi_{i}-\phi_{i+n}\right)\right\} \end{array}$$
where in Equation (Equation 9), there are two phase terms. The first term is the phase difference of interferometric phase after flat-earth phase removal, which expected value is 0. The second term is the differential phase caused by the RME, which can be obtained by extracting phase from Equation (Equation 9) meanwhile ignoring the first term phase.
(10)$$\begin{array}{cc} d\phi_{m} & =\arg\left\{{\langle I^{A}\cdot I^{B*}\rangle }_{m,m+1}\right\}\\ & \approx \phi_{m}-\phi_{m+1} \end{array}$$
where $\phi_{m}$ and $\phi_{m+1}$ are the integral equivalent phases of $\phi_{i}$ and $\phi_{i+n}$ at subaperture *m* and *m* + 1, Obviously, Equation (Equation 10) is the differential form of the RME phase. In particular, if the RME is a linear function, Equation (Equation 10) can be further written as
(11)$$d\phi_{i}=2\pi f_{d}\cdot \Delta t_{m}$$
where $\Delta t_{m}=n\cdot PRT$ is the time difference of the adjacent subaperture in the slow-time domain, *PRT* is the pulse repetition time, and $f_{d}$ is the linear proportionality coefficient of RME. Therefore, for high-order RME, the phase of the RME can be obtained directly by the RME differential phase integration of Equation (Equation 10). For linear RME, the linear coefficient of the RME can be obtained from Equation (Equation 11) and then linearly fitted to obtain the linear RME phase.

### 3.2. Investigation of RME Fitting Scheme in Stripmap and Spotlight SAR Modes

It should notice that the derivations of Equations (10) and (11) are designed for a single scattering point, and the imaging time is equal to the synthetic aperture time for a single scattering point. In practical applications, the SAR image is composed of the focused results of distributed scattering points. For different radar working modes, the correspondence between the imaging time of a single scattering point and the imaging time of the entire image is different. To make different scattering points have the same Doppler bandwidth in the same subaperture, it is more reasonable to divide the subapertures uniformly in the azimuth Doppler domain than to divide the subapertures in the slow-time domain. To accurately estimate RME under different SAR working modes, it is necessary to figure out the representation form of RME in SAR images under different working modes. The following analysis is performed to the difference between the typical high-resolution SAR modes, the stripmap mode, and the spotlight mode.

As shown in Figure 2, the scattering points $P_{1},P_{2},\ldots ,P_{K}$ are arranged in the azimuth direction. It is assumed that the imaging start pulse index of the scattering point $P_{k}\left(k=1,2,\ldots ,K\right)$ in the subaperture $m\left(m=1,2,\ldots ,M\right)$ is $s_{k,m}$, the end pulse index is $o_{k,m}$, and the differential interferometric phase of $P_{k}$ in subaperture *m* and subaperture *m* + 1 is $d\phi_{k,m}$. Thus, the imaging start pulse index of the entire image is $s_{I}=s_{1,1}$ and the end pulse index is $o_{I}=o_{K,M}$. At the same time, Equation $o_{k,m}=s_{k,m}+n=s_{k,m+1}$ is established. From Equation (Equation 10), the interferometric result through subaperture *m* and subaperture *m* + 1 can be written as
(12)$${\langle I^{A}\cdot I^{B*}\rangle }_{m,m+1}=\sum_{i=s_{k,m}}^{s_{k,m}+n}\tilde{C_{i}}\tilde{C_{i+n}}\exp\left\{j\left(\varphi_{i+\left(m-1\right)n}-\varphi_{i+mn}\right)\right\}\exp\left\{j\left(\phi_{i+\left(m-1\right)n}-\phi_{i+mn}\right)\right\}$$

To further analyze the similarities and differences between different types of RME in the stripmap SAR and spotlight SAR subaperture interferograms, it is necessary to discuss the following four cases.

(1)Assume InSAR works in spotlight mode and RME has a linear form, we have
(13)$$\begin{array}{c} s_{1,m}=s_{2,m}=\ldots =s_{K-1,m}=s_{K,m}\\ o_{1,m}=o_{2,m}=\ldots =o_{K-1,m}=o_{K,m}\\ d\phi_{1,m}=d\phi_{2,m}=\ldots =d\phi_{K-1,m}=d\phi_{K,m}\\ d\phi_{k,1}=d\phi_{k,2}=\ldots =d\phi_{k,M-1}=d\phi_{k,M} \end{array}$$
It can be illustrated that for the linear RME and SAR works in the spotlight mode, the differential phases of different scattering points in the same subaperture are the same, and the differential phase of the same scattering point in different subapertures is also the same. Therefore, the linear coefficient of the linear RME can be calculated by Equation (Equation 11), and the RME phase can be obtained by linear fitting.(2)Assume InSAR works in spotlight mode and RME has a high-order polynomial form, we have
(14)$$\begin{array}{c} s_{1,m}=s_{2,m}=\ldots =s_{K-1,m}=s_{K,m}\\ o_{1,m}=o_{2,m}=\ldots =o_{K-1,m}=o_{K,m}\\ d\phi_{1,m}=d\phi_{2,m}=\ldots =d\phi_{K-1,m}=d\phi_{K,m} \end{array}$$
It can be illustrated that for the high-order RME and SAR works in the spotlight mode, the differential phases of the different scattering points in the same subaperture are the same, which is the same as in the Case (1). However, unlike Case (1), the differential phase of the same scattering point in different subapertures is generally different. Since the time intervals of the subapertures in spotlight SAR are independent of each other, when the subaperture number is sufficient, the RME in each subaperture can be considered to be linear. Therefore, the linear coefficient of the linear RME in each subaperture can be estimated by Equation (Equation 11), and the RME phase of the full imaging time is obtained by multi-linear high-order fitting.(3)Assume InSAR works in stripmap mode and RME has a linear form, we have
(15)$$\begin{array}{c} s_{1,m}<s_{2,m}<\ldots <s_{K-1,m}<s_{K,m}\\ o_{1,m}<o_{2,m}<\ldots <o_{K-1,m}<o_{K,m}\\ s_{k,1}<s_{k+1,1}<o_{k,M}<o_{k+1,M}\\ d\phi_{1,m}=d\phi_{2,m}=\ldots =d\phi_{K-1,m}=d\phi_{K,m}\\ d\phi_{k,1}=d\phi_{k,2}=\ldots =d\phi_{k,M-1}=d\phi_{k,M} \end{array}$$
It can be illustrated that for the strip mode, there is a partial coincidence interval $\left[s_{k+1,1},o_{k,M}\right]$ in the imaging time interval of two adjacent scattering points $P_{k}$ and $P_{k+1}$ (the imaging time intervals are $\left[s_{k,1},o_{k,M}\right]$ and $\left[s_{k+1,1},o_{k+1,M}\right]$, respectively). When the RME is linear, since the subaperture is divided in the azimuth Doppler domain, the RME differential phases of the different scattering points in the same subaperture are the same. At the same time, since the subaperture is evenly divided, the RME differential phase of the same scattering point in different subapertures is also the same. Therefore, the linear coefficient of the linear RME can be calculated by Equation (Equation 11) as in Case (1) and Case (2), and the linear RME phase can be obtained by linear fitting.(4)Assume InSAR works in stripmap mode and RME has a high-order polynomial form, we have
(16)$$\begin{array}{c} s_{1,m}<s_{2,m}<\ldots <s_{K-1,m}<s_{K,m}\\ o_{1,m}<o_{2,m}<\ldots <o_{K-1,m}<o_{K,m}\\ s_{k,1}<s_{k+1,1}<o_{k,M}<o_{k+1,M} \end{array}$$
In this case, the latter two equations in (15) no longer hold. When the high-order RME acts on the stripmap mode SAR, the interferometric phases of the azimuth-ordered scattering points in the same subaperture reflect the RME differential phases in different time intervals and are continuous in the time domain. Therefore, the RME differential phase still appears as a high-order form along the azimuth direction, and it is not feasible to estimate the linear coefficient through the RME differential phase. At the same time, since the imaging time corresponding to the adjacent subapertures is partially coincident, it is necessary to obtain the RME differential phase of the full imaging time by sub-image differential interferometric phase in a “sliding window” splicing manner, and then the full imaging time RME phase can be obtain through the RME differential phase interpolation and integration.

In addition to the discussion of the above four cases, it should be noticed that the derivation of Equation (Equation 9) is derived in the case of a point target, and the actual SAR pulse compression signal contains many point target signals at the same range unit. It should be noted that the subaperture differential interferometric phase contains not only the differential interferometric phase of the RME, but also the noise phase caused by the image decoherence. In the BP-MSQ algorithm, the effect of noise phase on linear coefficient estimation can be eliminated effectively by averaging the differential interferometric phase. The effect of phase integration can also be attenuated by the smoothing of the differential phase curve in stripmap InSAR mode.

### 3.3. Detailed Algorithm Procedure

The main purpose of the above four cases is to adopt a more suitable error estimation method for different situations, the core idea of algorithm in different cases is same. The differential phase of the RME is firstly obtained from the sub-image interferometry of different oblique angles, and then the RME can be obtained through differential phase fitting. The backprojection multisquint algorithm for airborne InSAR RME estimation can be expressed more clearly and intuitively through the following process. The flow chart is shown in Figure 3.

Step (1)BP imaging using track information. Firstly, the track information can be extracted from IMU/GNSS. Then, the corresponding single-look complex image of each antenna can be obtained by BP algorithm from the echo data of the master antenna and the slave antenna.Step (2)Subapertures segmenting in azimuth Doppler domain. The Doppler spectrum of the single-look complex image of the master antenna and the slave antenna is uniformly divided into *M* frequency bands, and then multi-look images of the two antennas, i.e., a plurality of subaperture images are obtained.Step (3)Subaperture images differential interferometry. Firstly, the image of the subaperture with the same radar sight looking angle of the master antenna and the slave antenna is multiplied to obtain a subaperture interferogram. Then, the adjacent subaperture interferograms are multiplied by conjugate to obtain a subaperture differential interferogram.Step (4)RME estimation. The RME differential phase can be extracted from the subaperture differential interferogram, and then the corresponding strategy is adopted according to the SAR working mode and the RME type to estimate the value of the RME in the time domain over the entire imaging time.Step (5)BP imaging with RME compensation. BP algorithm was used to refocus the image and compensate RME at the same time to obtain the interferogram corrected by RME.

## 4. Experiments

To verify the accuracy and feasibility of the method described in this paper, several sets of simulation data are used to analyze the above theory, and the residual motion error estimation are performed base on the linear RME and high-order RME, respectively. To verify the applicability of the method for real measured InSAR data, the RME estimation is performed on a pair of measured data with large RME by the method described above. The measured data are provided by the National Lab of Radar Signal Processing and Collaborative Innovation Center of Information Sensing and Understanding of Xidian University. The computer platform is Windows7 (Microsoft Corporation, Redmond, WA, USA) 64-bit operating system, E5-2643@3.3 GHz CPU, 32 GB memory and MATLAB (MathWorks Corporation, Natick, MA, USA) Version R2014a.

### 4.1. Linear RME Simulation Experiment with Stripmap SAR

In this experiment, two sets of stripmap mode SAR echo data are simulated, in which one is without RME and the other is blurred with RME. The RME is set to be linear. After imaging with BP algorithm, the subaperture division is processed in the azimuth Doppler domain. The simulation scene settings are shown in Figure 4, and the simulation parameters are shown in Table 1.

As shown in Figure 4, Figure 4a is an imaging result of the simulation scene, and Figure 4b is a preset elevation information of the simulation scene. Where the simulation scene width is 256 m × 256 m, the maximum scene elevation is 45 m, the X direction grid spacing is 0.262 m, and the Y-direction grid spacing is 0.482 m. Figure 4a,b are the interferogram and coherence map of the complex image of the simulated scene without RME. It can be seen from Figure 4 that the imaging result of the simulation scene without RME is focused, the interferogram is clear. The coherence is 0.998 according to Figure 4d, which is close to the ideal value. Add linear RME to the above scene during simulation, and the simulation results are shown in Figure 5.

As shown in Figure 5, Figure 5a,b are simulation results when linear RME is added. Under the influence of linear RME, the interferogram exhibits linear modulation along the azimuth direction, and the coherence reduced from 0.998 to 0.969. Figure 5c is the RME differential phase distribution along the X direction after the adjacent sub-images interferometry by the number of subapertures (sublooks) *M* = 8. It is known from Figure 5c that the linear RME subaperture differential phase fluctuates around a fixed value along the X direction, which is consistent with the description of Case (3) in Section 3.2. The linear coefficient of RME can be calculated from the differential phase mean to estimate the linear RME phase. The comparison between estimation results of RME with the preset value is shown in Figure 5d. “BP-MSQ” indicates the estimation result of the algorithm proposed in this paper, and “Ori-MSQ” indicates the conventional multisquint technique mentioned in [2]. Under the BP-MSQ algorithm, the maximum error is 0.032 rad and the root mean square error(RMSE) is 0.018 rad. In Ori-MSQ, the maximum error is 0.194 rad and the RMSE is 0.097 rad. These results show that the BP-MSQ algorithm can estimate linear RME more accurately than conventional multisquint technique.

### 4.2. Simulation with High-Order RME

To further verify the similarities and differences of the algorithm described in different radar working modes, the spotlight SAR, and stripmap SAR echo data with high-order RME are simulated, respectively. After imaging with BP algorithm, the simulation parameters are also shown in Table 1. As shown, the RME is estimated based on the differential phase and using the corresponding RME data fitting method. The high-order RME function in the simulation is a cosine function defined as
(17)$$\begin{array}{c} {RME}_{high-order}\left(t\right)=0.64\cos\left(2\pi t\right)-0.36 \end{array}$$
where *t* is the azimuth time, the simulation results when InSAR works in spotlight mode and RME has a high-order form is shown in Figure 6.

As shown in Figure 6, Figure 6a,b indicate that the presence of the high-order RME causes the spotlight mode SAR interferogram to exhibit modulation characteristics in the azimuth direction, and the coherence is significantly reduced. Figure 6c is a differential phase distribution in the X direction (azimuth direction) when the number of subapertures (sublooks) *M* = 8. For the spotlight SAR, the subaperture differential interferometric phase fluctuates around a fixed value, which is consistent with the analysis of Case (2) in Section 3.2. The estimation result is shown in Figure 6d when the number of subapertures is *M* = 32, in which the maximum error is 0.029 rad and the RMSE is 0.015 rad compared to the preset value. The result indicates that the method is applicable for spotlight SAR with high-order RME.

However, it should be noted that the subaperture differential interferometric phase is the integral equivalent phase in the current subaperture and the accuracy of RME estimation depends on the number of subapertures. On the one hand, RME in each subaperture can be considered to be linear only when the number of subapertures is sufficient. On the other hand, the noise phase is strong when the subaperture is too much. Both of the above cases could lead to a large error for RME linear coefficient estimation in a subaperture. In the experiment, the maximum error is 0.074 rad and the RMSE is 0.041 rad when the number of subapertures is *M* = 16. Moreover, the maximum relative error is 0.067 rad and the RMSE is 0.024 rad when the number of subapertures is *M* = 64. The comparison results show that the above analysis is correct when InSAR works in spotlight mode and RME is a high-order function. Figure 7 shows the simulation results when the radar is working in stripmap mode and RME is a high-order function.

For stripmap mode SAR, as shown in Figure 7a,b, the interferogram and coherence map are also greatly affected by high-order RME. Unlike the spotlight mode, the RME differential phase exhibits a “sliding window” along the X direction, as shown in Figure 7c, which is consistent with the analysis of Case (4) in Section 3.2. The RME differential phase for entire imaging time can be obtained by “sliding window” splicing. Figure 7d is a comparison of the RME estimated phase with the preset value when the image is divided into the number of subapertures *M* = 16. The maximum error is 0.280 rad and the RMSE is 0.070 rad compared to the preset value. Compared with the piecewise linear RME estimation of the spotlight mode, the RME is obtained by integrating the differential phase in the strip mode so that the RME estimation result is more sensitive to the noise phase. By smoothing the differential phase curve, a more accurate RME estimation result can also be obtained.

The above simulation experiments show that the method described in this paper is consistent with the theoretical analysis in Section 3.2. Although the working modes of SAR are different and RME has different types, RME estimation can be performed using the backprojection multisquint algorithm, not only for stripmap SAR and spotlight SAR. It is necessary and beneficial to select a more suitable RME data fitting method in the actual application process.

### 4.3. Actual InSAR System Data Processing

To further verify the performance of the above method on the actual InSAR system, this experiment uses the measured data of airborne millimeter-wave InSAR to test the method mentioned in this paper. The InSAR system works in stripmap mode, and the system parameters are shown in Table 2. The BP algorithm is used to focus and analyze the characteristics of the interferogram before and after RME compensation. In addition, the RD algorithm is used for imaging to compare the performance of the two algorithms on the measured data.

As shown in Figure 8, the subgraph (a) is the BP imaging result of the experimental airborne millimeter SAR system. The terrain in the scene is relatively flat, and the subgraph (b) is the subgraph interferogram and coherence map before the RME compensation. The scene interferogram exhibits modulation characteristics along the azimuth direction. Theoretically, the phase spectrum distribution of the flat potential area should be relatively uniform. Subgraphs (d) and (e) are the RME subaperture differential phase and RME obtained from BP subaperture interferometry using BP-MSQ. Subgraph (c) is the interferogram and coherence map after RME compensation, in which the coefficient and interferogram are clear, and the average coherence is increased from 0.80 to 0.85. It is indicated that the BP-MSQ described in this paper is feasible and effective for the actual InSAR system.

Actual InSAR system data processing with RD algorithm is shown in Figure 9, in which the conventional “Two-step” MoCo method is applied. As shown in Figure 9, since the imaging projection plane of the RD algorithm is different from the imaging projection plane of the BP algorithm, comparing the graph (a) in Figure 8 and the graph (a) in Figure 9, the imaging results of the same scene region are different. Because the disadvantages of RD algorithm, large deformation scale, and serious decoherence, as shown in subgraphs (b) and (c), the interferogram is relatively uniform but the coherence is lower than that of the BP algorithm. The experimental results show that the BP algorithm is superior to the RD algorithm in terms of accurate error compensation.

## 5. Conclusions

For the RME refined estimation and correction of airborne InSAR, this paper proposes the BP-MSQ algorithm and demonstrates its availability and effectiveness in airborne InSAR RME estimation. First, the proposed BP-MSQ algorithm estimates polynomial parameters of RME through the subaperture differential phase, while it analyzes the representation of RME in the azimuth time domain, it is shown that BP-MSQ algorithm is effective for both linear RME and high-order RME. Second, the RME estimation algorithm is designed more flexible and adjustable by analyzing the RME estimation theory in different modes. Third, by avoiding the differential phase integration in RME estimation, the developed BP-MSQ outperforms the original MSQ in the case of strong noises.

## Figures and Tables

**Figure 1 sensors-19-02342-f001:**
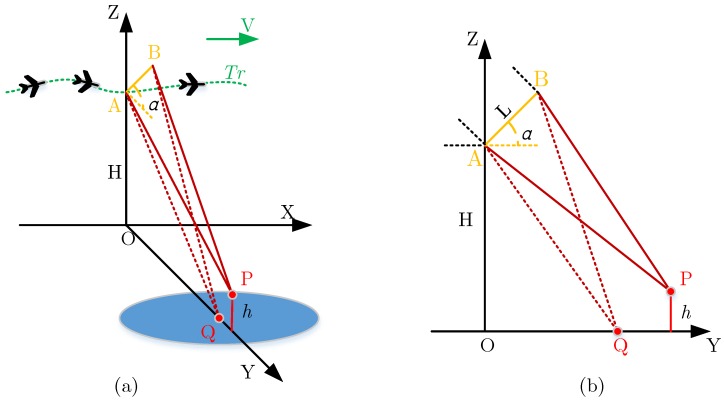
Airborne InSAR system model. (**a**) 3D diagram of InSAR (**b**) lateral view.

**Figure 2 sensors-19-02342-f002:**
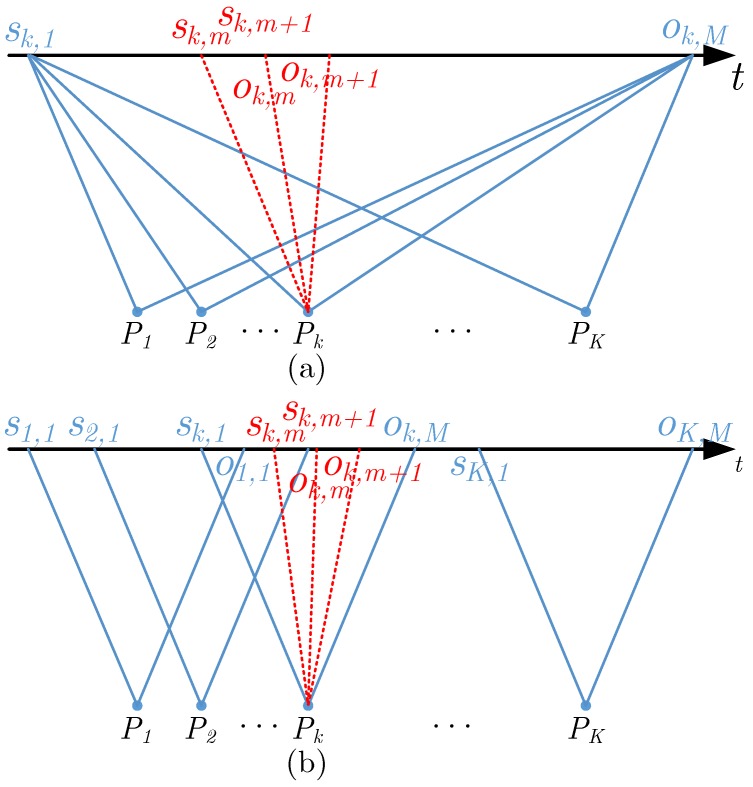
Schematic diagram of synthetic aperture for spotlight SAR and stripmap SAR. (**a**) Spotlight SAR (**b**) Stripmap SAR.

**Figure 3 sensors-19-02342-f003:**
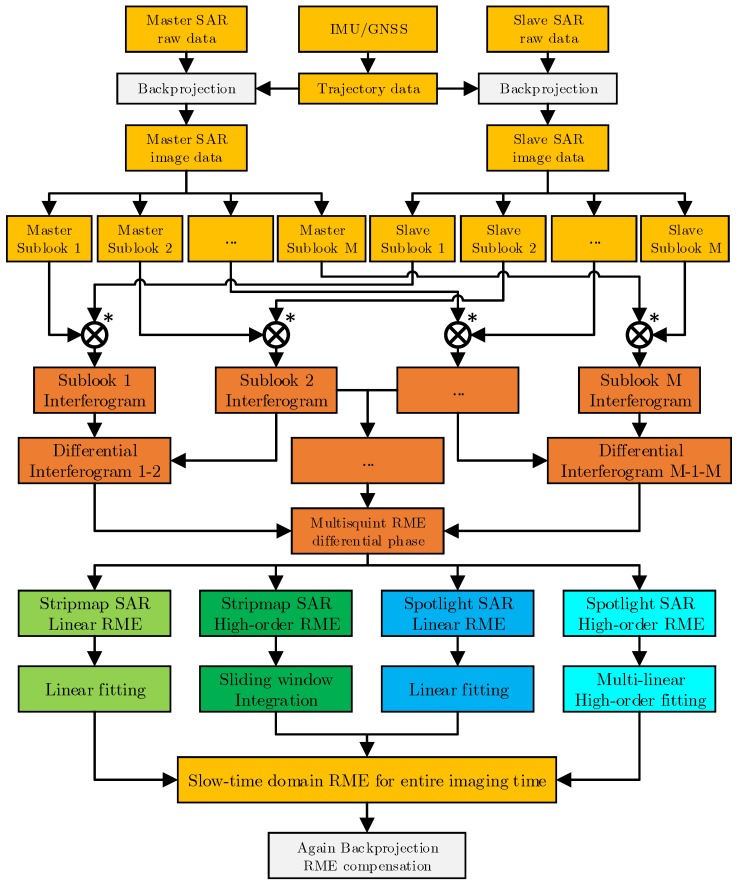
Flowchart of proposed BP-MSQ algorithm.

**Figure 4 sensors-19-02342-f004:**
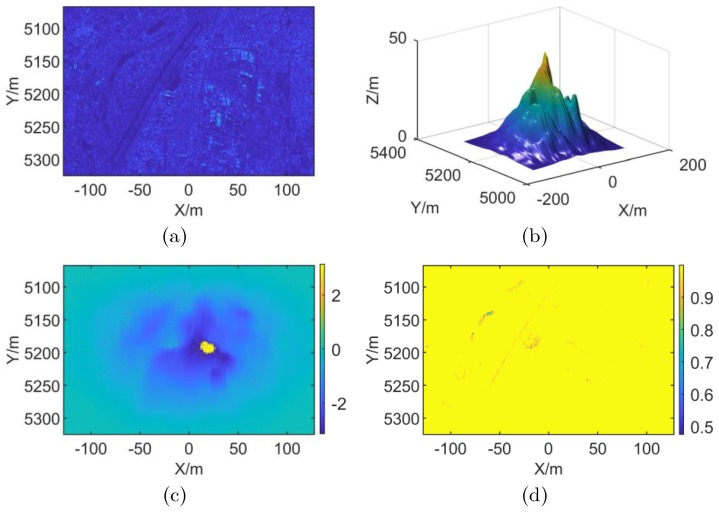
Simulation scene information. (**a**) Simulation Scene (**b**) Preset DEM (**c**) Interferometric phase without RME (**d**) Coherence map without RME.

**Figure 5 sensors-19-02342-f005:**
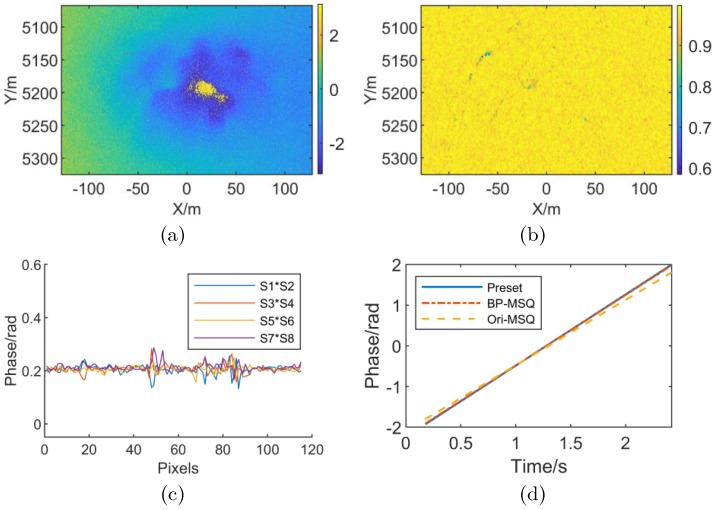
Stripmap SAR simulation results with linear RME. (**a**) Interferometric phase with linear RME; (**b**) Coherence map with linear RME; (**c**) Subaperture differential interferometric phase (’S1’ denotes the subaperture 1, and so forth); (**d**) Comparison of preset value and estimation result of RME phase.

**Figure 6 sensors-19-02342-f006:**
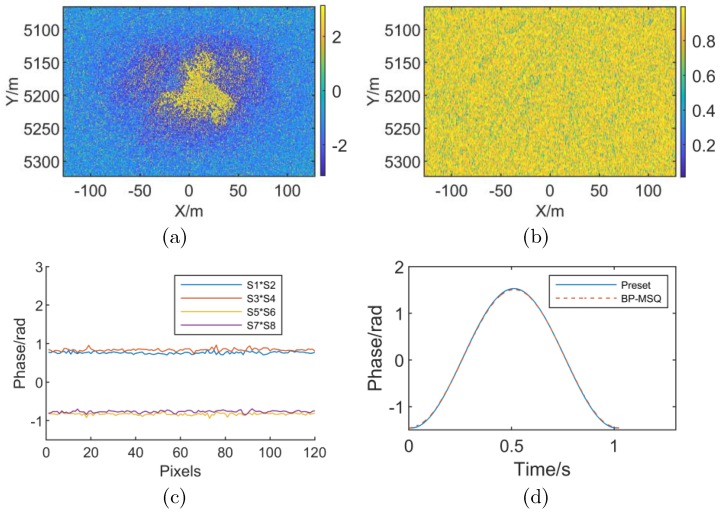
Spotlight SAR simulation results with high-order RME. (**a**) Interferometric phase with high-order RME (**b**) Coherence map with high-order RME (**c**) Subaperture differential interferometric phase (**d**) Comparison of preset value and estimation result of RME phase.

**Figure 7 sensors-19-02342-f007:**
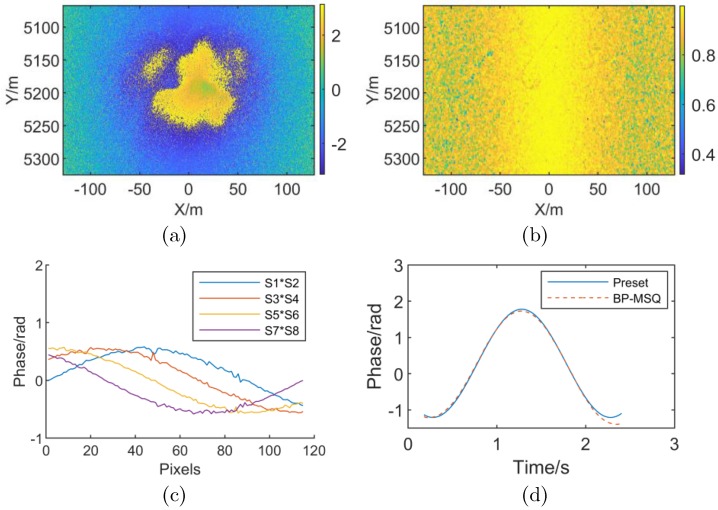
Stripmap SAR simulation results with high-order RME. (**a**) Interferometric phase with high-order RME (**b**) Coherence map with high-order RME (**c**) Subaperture differential interferometric phase (**d**) Comparison of preset value and estimation result of RME phase.

**Figure 8 sensors-19-02342-f008:**
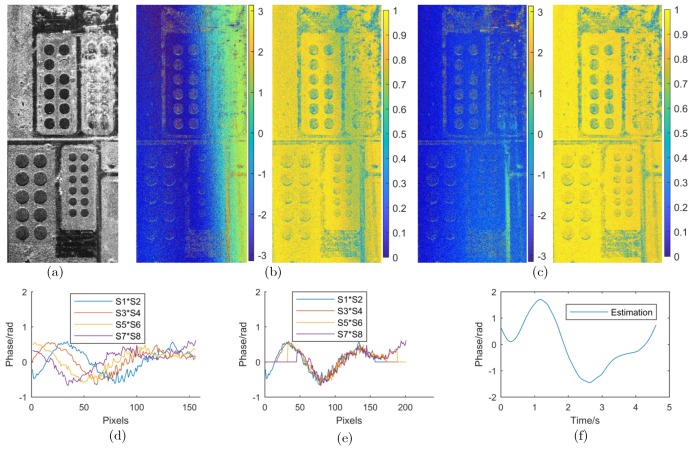
Actual InSAR system data processing with BP algorithm. (**a**) Imaging result of Scene with BP algorithm; (**b**) Interferometric phase and coherence map before RME compensation; (**c**) Interferometric phase and coherence map after RME compensation; (**d**) Subaperture differential interferometric phase; (**e**) Subaperture differential interferometric phase after ”sliding window”; (**f**) RME phase estimation.

**Figure 9 sensors-19-02342-f009:**
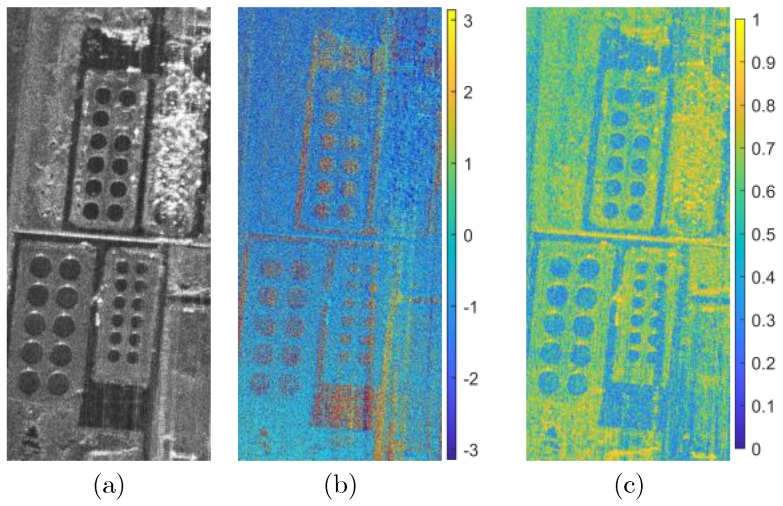
Actual InSAR system data processing with RD algorithm. (**a**) Imaging result of Scene with RD algorithm; (**b**) Interferometric phase after MoCo; (**c**) Coherence map.

**Table 1 sensors-19-02342-t001:** Parameters of simulate InSAR system.

Items	Symbol	Value
Wave Length	λ/(mm)	18
Band Width	B/(MHz)	150
Flight Speed	V/(m/s)	200
Pulse Repetition Frequency	PRF/(Hz)	2000
Flight Height	H/(m)	3000
Base Line	L/(m)	1.21
Baseline Obliquity	α/(${}^{\circ }$)	45

**Table 2 sensors-19-02342-t002:** Parameters of actual InSAR system.

Items	Symbol	Value
Wave Length	λ/(mm)	8.57
Band Width	B/(MHz)	900
Flight Speed	V/(m/s)	100
Pulse Repetition Frequency	PRF/(Hz)	5000
Flight Height	H/(m)	3000
Base Line	L/(m)	0.087
Baseline Obliquity	α/(${}^{\circ }$)	45

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
