# Peer review of "Residual Motion Error Correction with Backprojection Multisquint Algorithm for Airborne Synthetic Aperture Radar Interferometry"

_sensors, 2019, doi:10.3390/s19102342_

Round 1

Reviewer 1 Report

This paper proposes a novel motion error for airborne bistatic interferometric SAR. The proposal itself is interesting however; the paper lacks numerical comparison between proposed and conventional method.

1. In experiments, the authors have to perform comparison among the proposed and the conventional method(s). Motion error estimation for airborne (interferometric) SAR has various preceding researches and some of them are based on back projection method.

2. Write the simulation condition. What is the exact function of "high-order RME"?

3. Please evaluate the performance (e.g., comparison between actual and estimated RME) numerically.

4. In L. 84, the authors wrote that the slave antenna is only for receiving, which means that this is a bistatic system. Please clarify it in Section 1 too.

5. The term "Look" in Fig. 3 seems to be Subaperture or Sublook.

6. In Fig. 4, "with no RME" is misleading. "without RME" is better.

The scale of interferogram is not shown in this article.

7. Please specify the amount of RME, number of multilook and the number of subaperture first. Is there any dependency on the number of subaperture for RME estimation?

8. Is "Moco" in Fig. 9 "MoCo"? How or what method did the authors apply? Please specify.

9. Minor corrections

9.1. It is better to put the proposal (current Section 3.3) first in Section 3 to clarify the system.

9.2. "i_m +n (+2n)" in Eq. (8) seems "s_m +n (+2n)" while the range of summation in Eq. (9) is i=s_m:s_m+n. Please revise them.

9.3. The term “Doppler” starts from capital “D”.

9.4. In L. 83, is “mater” master?

Author Response

Thank you for your valuable suggestions, our responses have been listed in the PDF file.

Reviewer 2 Report

[1] Grammar and writing style need to be improved, professional editing service is recommended.

[2] Lines 58-74: The major contributions should be summarized in the Conclusion.

[3] Eqn.(2): Please elaborate how the first expression is derived.

[4] Eqns.(8) and (9): Please double check the upper bound of summation.

[5] Figures 4-7: Please insert color bar for interferometric phase in each subfigure (a).

[6] Line 261: Please elaborate why the coherence in Figure 5(b) is sharply reduced as compared with Figure 4(d). They do not look very different.

Author Response

(The authors gave the same response as above.)

Reviewer 3 Report

The paper has all merits to be accepted for publication after minor revision.

Author Response

(The authors gave the same response as above.)

Round 2

Reviewer 1 Report

The revised paper seems fine to be published.

This manuscript is a resubmission of an earlier submission. The following is a list of the peer review reports and author responses from that submission.

Round 1

Reviewer 1 Report

I don’t think this manuscript is ready to be submitted to a journal for review. The language needs to be thoroughly polished. Many basic terms and well-known concepts in the manuscript are not correctly written. I am sorry to say I didn’t finish reading the manuscript for obvious reasons.

Here, I list a few examples about the language or errors in the manuscript, but there are too many to list. At least, this provides the authors some guidance on how to revise the manuscript.

You shouldn’t use ‘blend’. Instead you can use ‘combine’

Fig. 1 is not a good figure and needs to be revised. Baseline is mostly perpendicular to the flight path.

In equation 1, when you use ‘where’ to explain the variables, you shouldn’t do it in a new paragraph. What exactly does fc mean? There are different understandings of ‘center frequency’ in synthetic aperture radar.  I guess here you are meaning the radar center frequency, which corresponds to the radar wavelength. Also in the same place but above equation (1), ‘range-oriented focus through pulse compression’ should be simply ‘range focusing’.